# Mind the Gap! Reconciling Environmental Water Requirements with Scarcity in the Murray–Darling Basin, Australia

**Matthew J. Colloff * and Jamie Pittock**

Fenner School of Environment and Society, Australian National University, Canberra, ACT 2601, Australia; jamie.pittock@anu.edu.au
* Correspondence: Matthew.Colloff@anu.edu.au

**Abstract:** The Murray–Darling Basin Plan is a $AU 13 billion program to return water from irrigation use to the environment. Central to the success of the Plan, commenced in 2012, is the implementation of an Environmentally Sustainable Level of Take (ESLT) and a Sustainable Diversion Limit (SDL) on the volume of water that can be taken for consumptive use. Under the enabling legislation, the *Water Act* (2007), the ESLT and SDL must be set by the "best available science." In 2009, the volume of water to maintain wetlands and rivers of the Basin was estimated at 3000–7600 GL per year. Since then, there has been a steady step-down in this volume to 2075 GL year due to repeated policy adjustments, including "supply measures projects," building of infrastructure to obtain the same environmental outcomes with less water. Since implementation of the Plan, return of water to the environment is falling far short of targets. The gap between the volume required to maintain wetlands and rivers and what is available is increasing with climate change and other risks, but the Plan makes no direct allowance for climate change. We present policy options that address the need to adapt to less water and re-frame the decision context from contestation between water for irrigation versus the environment. Options include best use of water for adaptation and structural adjustment packages for irrigation communities integrated with environmental triage of those wetlands likely to transition to dryland ecosystems under climate change.

**Keywords:** water reform policy; wetlands; governance; environmental flows; climate adaptation; water politics; adaptation pathways

## 1. Introduction

The Murray–Darling Basin in south-eastern Australia (hereafter called the Basin) consists of the Baaka/Darling River and its tributaries in the north, characterised by highly variable inflows and mostly summer rainfall, and the Murray and Murrumbidgee rivers and their tributaries in the south, with more regular winter-spring inflows from snowmelt from the Great Dividing Range (Figure 1). From the latter 19th century, irrigation schemes were developed, with construction of large headwater dams in the early-mid 20th century. Prior to irrigation, much of the Basin consisted of pastoral leases of Crown (government) land, and before the 1840s, the land and waters were under Indigenous tenure and management ([1], pp. 53–85). By the 1950s, irrigation diversions and storage capacity were rising markedly with the post-war boom in irrigation, reaching a peak during the 1980s and 90s ([2], Figure 1 therein; [3], p. 270). Weirs were constructed to provide storage for town water supplies and mitigate flooding. In just over a century, the Basin had been transformed from natural, free-flowing rivers with large terminal wetlands that were the country of many Indigenous Nations and abounded with waterbirds, fishes, and other wildlife to a system of highly regulated rivers, irrigation districts, and regional towns, with major negative impacts on the ecological condition of rivers and wetlands [4,5].

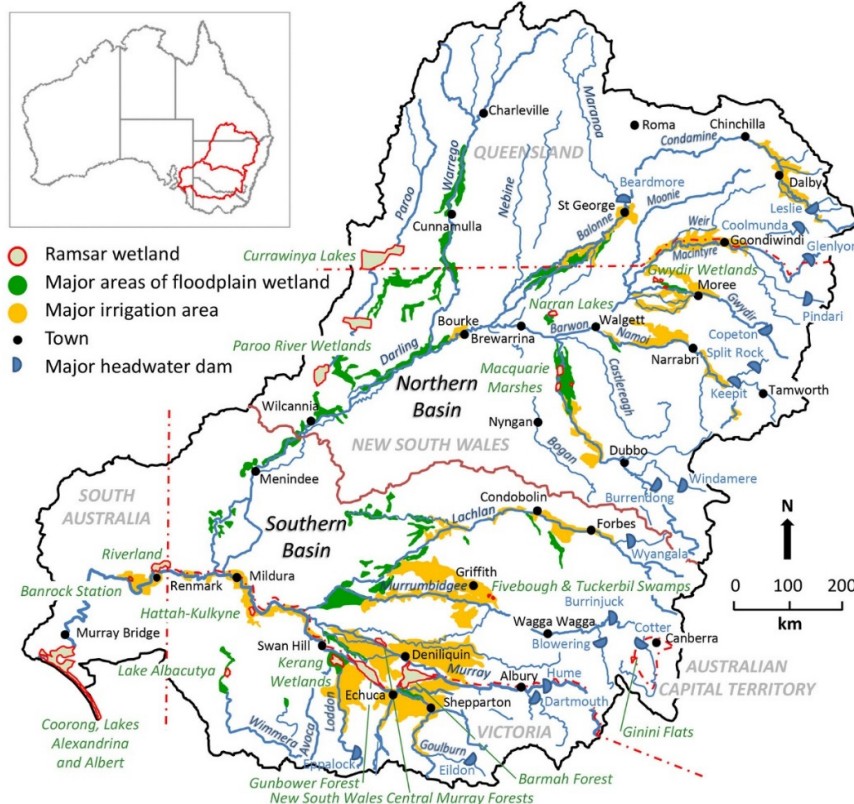

**Figure 1.** The Murray–Darling Basin showing major tributaries of northern and southern basins and proximity between major wetlands, floodplains, and irrigation districts. Note major headwater dams on most rivers, which supply irrigation water and environmental flows and major irrigation districts upstream of important wetlands. Inset: map of Australia showing the location of the Murray–Darling Basin.

Until recently, water resources were controlled by each Basin State and Territory, mostly for consumptive use. By the 1990s, irrigation water was grossly over-allocated, accounting for almost half of available surface water ([6], p. 32). Risks from over-allocation led to a cap on diversions in 1995 (the Murray–Darling Basin Cap) and greater inter-jurisdictional collaboration (so-called "co-operative federalism") [7], leading to the Council of Australian Governments (CoAG) reforms from 1994 [8] and the National Water Initiative (NWI) in 2004. These changes set the scene for the *Water Act* (Cth. 2007) and the Basin Plan (2012), reforms made more urgent by the Millennium Drought (1997–2010), among the longest and most severe in recorded history. The CoAG reforms and the NWI [9] remain the basis for current policies.

The Murray–Darling Basin Plan, a $AU 13 billion program to return water from irrigation to the environment, has been lauded as world's best-practice in water management [8,10,11]. Central to the Plan is the implementation of an Environmentally Sustainable Level of Take (ESLT) and a Sustainable Diversion Limit (SDL) on water for consumptive use, to be determined by the Murray–Darling Basin Authority (MDBA), the agency responsible for the design and implementation of the Basin Plan. Under the enabling legislation, the *Water Act*, the ESLT and SDL must be set by the "best available science." In 2009, water to maintain wetlands and rivers was estimated at 3000–7600 GL per year in the *Guide to the Basin Plan* [3]. Since then, this volume has been steadily stepped down due to repeated policy interventions due to droughts, concerns over water insecurity, and political contestation over negative impacts on irrigation communities.

These adjustments include "supply measures projects," most of which involve infrastructure to obtain the same environmental outcomes with less water. Yet, the average

annual volume of environmental water released under the Basin Plan is already falling far short of targets. The gap between what is required to maintain wetlands and rivers and what has been released is increasing under climate change. There has been a marked fall in Basin inflows in the last 20 years [12]; yet, the Basin Plan contains no direct allowance for climate change

We examine issues central to implementation of the Basin Plan: (1) *the step-down effect*: the gap between the volume of environmental water required and what Basin governments are prepared to allocate and use; (2) *the climate change gap* between water available when the Basin Plan was first developed in 2009–10 and what is available under climate change; (3) *the upstream-downstream gap*, where irrigation diversions have major negative effects on downstream communities and ecosystems, particularly in the northern Basin; and (4) *the implementation gap* between the objectives of the *Water Act* and their reluctant implementation by some Basin governments and the consequent undermining of trust. We present policy options that address the need to adapt to less water and re-frame the decision context from the contestation between water for irrigation versus the environment to water justice and "water for adaptation." These options include best use of water for adaptation and structural adjustment for irrigation communities integrated with environmental triage of those wetlands likely to transition to dryland ecosystems under climate change.

## 2. The Step-Down Effect

### 2.1. Environmental Water Allocations and Targets

The original estimated volume of additional water to be returned to the environment by the end of the first iteration of the Basin Plan in 2024, 3000–7600 GL/year, was a 22–56% reduction from the 13,623 GL historically allocated for irrigation [3]. This environmental water requirement must be determined by the "best available science" under the *Water Act* ([13], S21(4)(b)). In the ensuing outcry over the negative effect this large reduction would have on irrigation communities, the Chair of the MDBA, Mike Taylor, resigned and was replaced in 2011 by former New South Wales (NSW) politician and lobbyist, Craig Knowles, who travelled the Basin negotiating with irrigators, making it public that he had "a poor opinion of the *Guide*" and stated the Basin Plan "will contain our best estimates of the sustainable diversion limits and the environmentally sustainable level of take" [14]. The volume was then reduced to 2800 GL/year with no justification or rationale (Figure 2). The determination of the ESLT and SDL was a political fix not based on science and was thus unlawful [15]. Coming from the Chair of the Authority responsible for the Basin Plan, this step-down was more than just a win for irrigators. It set a precedent for a policy culture and context based not on the implementation of the objectives of the *Water Act* but their evasion, delay, and distortion. Most importantly, it broke trust with Basin communities by sending a message that those implementing the Basin Plan were above the law. That breach of trust has had far-reaching negative consequences for the MDBA and other government agencies ever since [16].

Following revised modelling, a further 50 GL was removed, followed by a 70 GL deduction after the northern Basin review [17]. In September 2015, the *Water Amendment Bill* (Cth.) limited water buybacks from irrigators to 1500 GL. However, only 1165 GL had been purchased, requiring water to be sourced by other means. Accordingly, in 2017, the MDBA determined that 605 GL (so-called "down water") could be made available for consumptive use via an increase in the SDL by implementing 36 "supply measures" projects under the Sustainable Diversion Adjustment Mechanism (SDLAM) [18,19]. The rationale is that the same environmental benefits could be achieved with less water through "environmental works and measures", "constraints relaxation projects" to remove flow restrictions, major restructuring projects (e.g., at Menindee Lakes and Yanco Creek) to offset large volumes of water by reducing evaporation and seepage, and "rules-based projects" to adjust river operations for environmental benefits [18,19]. These projects need to

be implemented by 2024, with a reconciliation between the original volume offset against the actual volume-equivalent implemented. Any shortfall requires the purchase of additional environmental water entitlements by the Commonwealth.

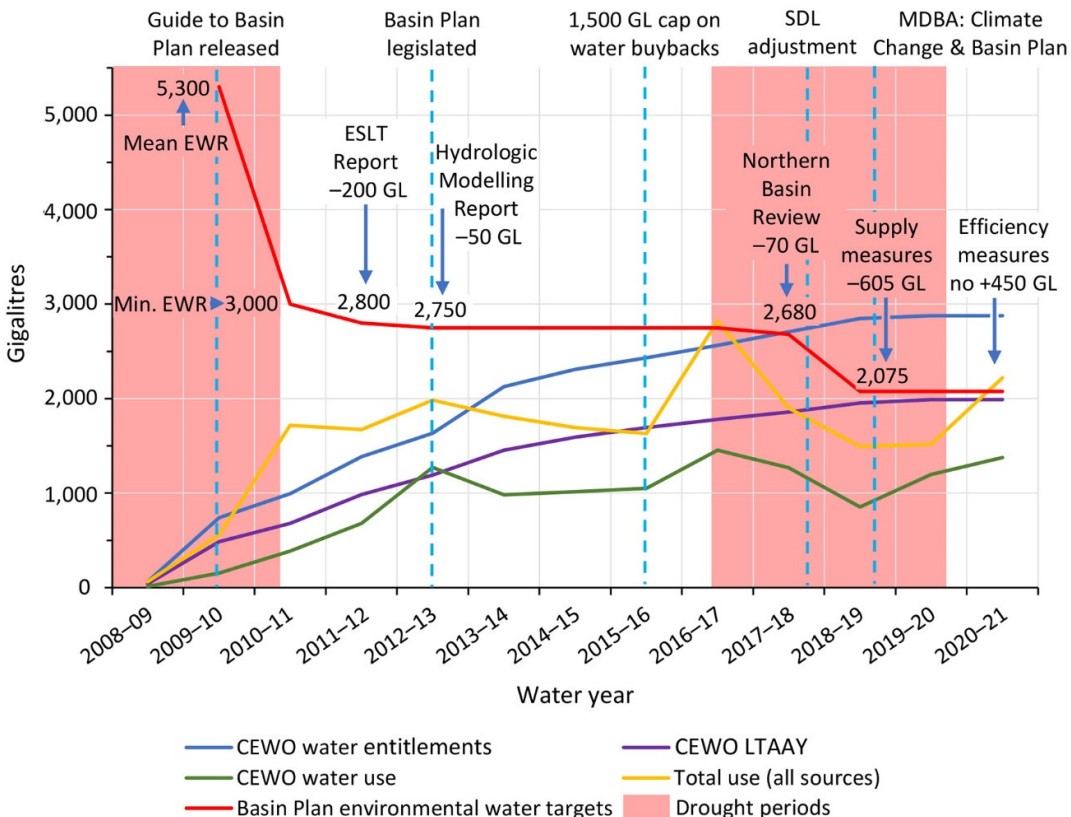

**Figure 2.** Recovery and use of water for the environment in the Murray–Darling Basin and adjustments to targets for environmental water requirements (EWR) in terms of long-term average annual yield (LTTAY) of Commonwealth Environmental Water Office (CEWO) water entitlements. Note that entitlement volumes are markedly higher than actual water recovered for the environment, which is indicated by the LTAAY, because some of these entitlements are of low security (including from overland flows) and unlikely to be realised except during periods of very high rainfall. ESLT, ecologically sustainable limit of take; EWR, environmental water requirements; MDBA, Murray–Darling Basin Authority; SDL, Sustainable Diversion Limit. Data on CEWO water entitlements and LTAAY from [20]. Data on environmental water use 2012–13 to 2018–19 from [21] and pre-2012–13 and post-2018–19 water use from [22] and State environmental watering annual reports (cf. Supplementary Material, references used to compile Table S1).

An assumption underpinning the SDLAM is that an additional 450 GL of environmental water (so-called "up water") would be available from irrigation efficiency projects by 2024. However, these projects are subject to a highly restrictive test of socio-economic neutrality. The Basin Plan sets limits to the adjustment of the SDL by five per cent (543 GL, based on an SDL of 10,873 GL at the time of the determination). Thus, a further 62 GL of additional water savings through efficiency projects is required to pass the five per cent rule. By June 2021, 2107 GL of surface water entitlements had been recovered but only 1.9 GL in efficiency measures. Following the amendments to the Basin Plan, the target for water recovery was 2075 GL per year plus 450 GL per year from efficiency projects [19]. It is clear that the stepping down of environmental water recovery was a strategy to fit targets to the trajectory of the volume recovered (Figure 2), thus reducing negative impacts on irrigators. It had nothing to do with actual environmental water requirements.

The targets for SDLAM and efficiency projects are likely to fall short. In 2016, then federal water minister Barnaby Joyce wrote to the South Australian water minister reneging on the delivery of the 450 GL of upwater [23] though in 2017, his successor committed to its delivery [24]. In February 2021, the on-farm Water Efficiency Program was abandoned as ineffective and too costly [25], to be replaced by unspecified off-farm projects, such as upgrades to irrigation districts, at grossly elevated cost of water recovery [26]. Following community concerns, the major restructuring projects at Menindee Lakes and Yanco Creek have been re-scoped but will likely yield much lower water savings than first envisaged. In June 2021, at the *River Reflections* Conference, hosted by the MDBA, Chief Executive Philip Glyde stated "It's really clear. Unless things change markedly, there's not going to be 605 gigalitres of projects completed by 2024…And nor is there going to be 450 GL worth of on-farm and off-farm efficiencies" [27]. Far from demonstrating "the flexibility and adaptability of the Plan" ([18], p. i), the SDLAM and efficiency projects have demonstrated their inherent weakness through the failure of co-operative federalism on water reform.

*2.2. Environmental Water Use—Triage by Default*

About 63% of the water recovered for the environment is held by the Office of the Commonwealth Environmental Water Holder (CEWO; the agency responsible for planning and managing the delivery of water for the environment under the *Water Act*); 13% by State agencies in New South Wales, Victoria, and South Australia; and 24% in other accounts, including The Living Murray program ([21], Figure 2 therein). In Queensland, environmental water is protected in catchment water resource plans via access rules rather than specific allocations. The CEWO partners with States and other water holders to co-ordinate the delivery of environmental water.

The 2020 Basin Plan Evaluation report claims environmental watering since 2012 "is having a significant and positive impact on the Basin environment" ([12], p. xiii). There is reason to be sceptical of this claim following the catastrophic fish kills at Menindee during the summer of 2018–19 [28–30]. Major shortfalls in restoring river flows have been reported, with annual flows 22% less than expected on the River Murray at the South Australian border [31]. The average volume of environmental water released (2012–13 to 2018–19) was estimated at 1905 GL/year [21], updated to 1897 GL/year with 2019–20 and 2020–21 included (Figure 2; Supplementary Material Table S1). The volume released between 2008–09 and 2012–13 averaged 1315 GL/year (Figure 2). Only 7% of the wetland area in 9/19 catchments received effective environmental water from 2014–15 to 2018–19 and only 0.8% of the area of all major wetlands. Some 20% was delivered to wetlands on the floodplain, with the rest for in-channel flows, and 89% of water was delivered in the southern Basin, mostly in the Murray and Murrumbidgee catchments. Under the Basin Plan, wetlands have not received the water they need, and the objectives for floodplain vegetation in the Basin-wide environmental watering strategy ([32], Appendix 2 therein) cannot be met by completion of the Plan in 2024.

The marked difference between the area within scope for environmental watering under the Basin Plan and what can actually be achieved indicates a process of what we refer to as "triage by default," whereby mostly smaller wetlands and the lower-lying, easily accessible parts of large wetlands can be flooded with a frequency to maintain their ecological character [21].

## 3. The Climate Change Gap

Under global warming, higher surface temperature drives greater evaporation leading to decreased runoff and inflows to rivers, even in regions where rainfall may be greater or the same [33,34]. Lower surface water availability will intensify demand and disputes over water access and use. Droughts are predicted to increase in frequency, duration, and severity [35]. The Millennium Drought ended in late 2010 with a strong La Niña event. Seven years later, runoff had not recovered to pre-drought levels in 37% of

Victorian catchments, indicating persistence of hydrological drought long after meteorological drought ceased [36].

The CSIRO Sustainable Yields Audit, though slightly dated, contains the most comprehensive projections of changes in rainfall and runoff for the Basin at catchment scale. Under a median warming scenario for 2030, projected reduction in mean annual runoff is 9% Basin-wide, 5–10% in the north-east and south, and 15% in the far south ([6], p. 24). Under a "dry" scenario (+1.6 °C above that of 1980 by 2030), decline in mean annual runoff is projected at up to 33%.

The effects of climate change were not accounted for in setting ESLTs and SDLs in the Basin Plan [37], which was unlawful under the *Water Act* [15]. The 2020 Basin Plan Evaluation states, "At the time of the Basin Plan's development, the CSIRO advised the MDBA that while climate change was a known risk, the Basin Plan should use the longest possible climate record for hydrologic modelling to encapsulate a range of climate conditions (Chiew et al., 2009). Guided by this advice, the 114-year climate history (1895–2009) was used as the climate baseline for the Basin Plan modelling" ([12], p. 22). This statement is demonstrably false. The CSIRO advice was as follows: "The climate sequence used for modelling over the period of implementation of the first Basin Plan (next 10–15 years) should be based on scenarios ranging from the recent climate over the past 10–20 years (a very dry scenario although drier conditions are possible) and future climate scenarios obtained using the daily scaling method described" ([38], p. 3). MDBA rejected this advice because modelling recent climate would have resulted in lower estimates of water availability and hence lower SDLs, which would have been politically harder to implement.

The effects of climate change are happening now in the Basin, especially increased temperature and the number of extreme hot days per year. However, perceptions of declining inflows and periods of no flow, such as occurred in the Baaka/Darling River prior to the 2018–19 fish kills at Menindee, have sometimes been attributed solely to drought and climate change by governments and their agencies, ignoring the effects of irrigation diversions [12,30]. The 2020 Basin Plan Evaluation main report claims annual inflows to the River Murray have declined by 39%: from 11,234 GL between 1895–2000 to 6841 GL between 2000–2020 [12]. The MDBA arrived at this claim by "cherry picking" the data: calculating the average for the two periods and deducting the latter from the former. This crude arithmetic ignores the fact the data represent a time series and thus will show temporal autocorrelation, which has to be accounted for before statistical inference can be drawn. In addition, the period 2000–2020 was in drought for 70% of the time compared with 35% for 1895–2000. This is hardly the "best available science." Regardless, the data have been accepted without question in at least one publication [39]. By the same selective method, inflows during dry periods between 1895–1915 and 1929–1949 would be deemed almost as low, with mean inflows of 7476 GL and 8527 GL/year, respectively.

The origin of the inflow data is not stated in the main report, but it appears to be from the Source Murray Model. However, the 2020 Hydrological Analysis Evidence Report is based not on this data but on inflows since 1911 from the Bureau of Meteorology Australian Water Resources Assessment Landscape model 6.0 (AWRA-L) available at ref. [40]. The hydrology evidence report states that "the AWRA-L runoff data provided a valuable proxy for annual climate conditions across each catchment" ([41], p. 15). To add to the confusion, the Murray inflows data in the main report does not match Murray inflows provided by MDBA in their answer to a question on notice from the Senate Rural and Regional Affairs and Transport Legislation Committee [42]. Figure 3 shows the Murray inflow data compared with the inflow data for the southern Basin, with drought periods mapped on, indicating clear inter-decadal cycles of wet and dry years. Ascribing an apparent recent reduction in inflows solely to climate change is incorrect. Changes in inflows need to be partitioned between the effects of climate change, inflow interception activities, and the magnitude of irrigation diversions, particularly from floodplain harvesting, which we outline below.

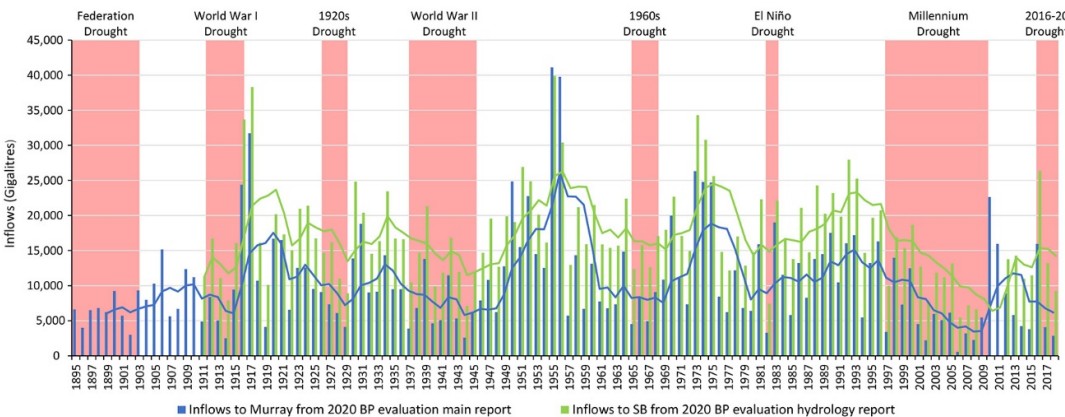

**Figure 3.** Inflows to the Murray and southern Basin (SB) from the *2020 Basin-Plan Evaluation Report* [12] and the *2020 Basin Plan Evaluation Hydrological Analysis Evidence Report* [41] and data dashboard [40], with five-year moving averages and the eight periods of major drought between 1895–2018 (data from [43]). The average inter-drought period is 9 years 11 months. Note the cyclic inter-decadal variability between drought and wetter periods and also that data is missing for the years 2009–2011 from the hydrological analysis evidence report.

## 4. The Upstream-Downstream Gap

### 4.1. Risks to Shared Water Resources

As water scarcity has increased in the Basin, farmers and others have begun adapting land and water use to climate change. The *Water Act* (S22(1)(3)) requires the Basin Plan addresses risks to the condition and availability of water resources arising from interception activities, climate change, changes to land use, and knowledge limitations. In 2006, CSIRO estimated reduction of river flows due to climate change, capture in farm dams, reduced flows due to improved irrigation efficiency, groundwater extraction, and increased transpiration from afforestation and regrowth after fires [44]. The MDBA investigated these risks to shared water resources extensively between 2008–2012 but did not address them explicitly in the Basin Plan, and the reports are not all publicly available.

It is clear these risks pose major threats to water availability. For example, floodplain harvesting take in the northern Basin is significant, increasing, and unregulated (cf. below). While 2750 GL of surface water was to be re-allocated to the environment under the Basin Plan, it permits increased groundwater take by 1548 GL/year (from 1786–3334 GL/year) [45] despite the risk of lower river base flows. No science to justify increased groundwater take has ever been published. Further, the $AU 3.5 billion program to subsidise on-farm irrigation efficiency improvements and return 700 GL/year of water for the environment may have recovered up to 630 GL/year less than expected due to the failure to account for a reduction in return flows [46]. MDBA consultants concluded only 121 GL/year of return flows were unaccounted for [47], but the MDBA continue to assert that any reduction in return flows is not a significant issue.

Risks to shared water resources were quantified by the Wentworth Group of Concerned Scientists using a method, agreed with the MDBA, to assess differences between observed river flows and those expected from MDBA hydrological modelling. Adjustments were made for environmental water and annual climate variability. Between 2012–19 and 2018–19, covering seven years since Basin Plan commencement, 22% of the expected water was missing from the River Murray at the South Australian border [31]. While climate was accounted for, other possible reasons for the deficit include inaccurate hydrological models, unregulated water take, double counting of return flows, and inflow interception activities. This gap of more than a fifth of expected flows raises serious questions over the rigour of Basin governance and the ability to meet environmental watering targets for flow-dependent ecosystems.

## 4.2. Irrigation Water Use and Floodplain Harvesting

Under the Basin Plan, changes to allocations of water for irrigation and the environment are due to come into effect only when State Water Resource Plans (WRPs) are fully operational by 2024. Accordingly, there has been no discernible reduction in irrigation water use since Basin Plan implementation (Figure 4). Irrigation water use and irrigated area track changes in rainfall, lagged by about a year, with declines during droughts and increases during wetter periods. It is important to note the volume of environmental water use is not currently contingent upon decreases in irrigation water use. CEWO water entitlements exceed the current environmental water target of 2075 GL/year and the long-term average annual yield (LTAAY) from those entitlements almost matches it (Figure 2). However, environmental water releases by CEWO are limited by operating rules and constraints [21,48] and track well below LTAAY.

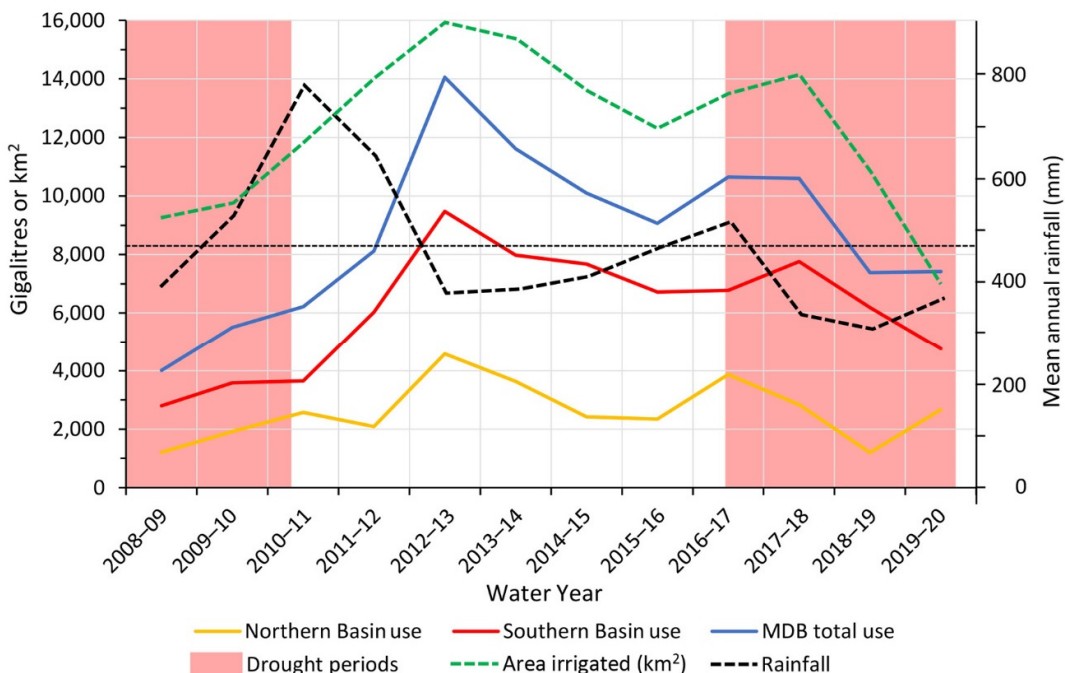

**Figure 4.** Irrigation water use in the Murray–Darling Basin from 2007–08 (when the *Water Act* received assent) to 2020–21 (solid lines; for northern and southern Basin and total), compared with the annual area irrigated and rainfall (black dashed line; long-term mean of 464 mm shown as black horizontal line). Data on irrigation water use from MDBA water audit monitoring reports on Cap implementation (1994–95 to 2011–12) [49] and transitional SDL water take reports (2012–13 to 2018–19) [50]. Data on area under irrigation from [51]. Data on mean annual rainfall for the Basin from Bureau of Meteorology annual national water accounts [20].

The negative effects of irrigation diversions on downstream communities and ecosystems perpetuates injustices between those with control over water access and those without. The upstream-downstream gap is prominent in the northern Basin, with major cotton growing districts along the Barwon–Darling, Condamine–Balonne, Macintyre, Gwydir, Namoi, and Macquarie rivers where floodplain water harvesting is widely practiced (Figure 1). These regions are upstream of the Ramsar wetlands of Narran Lakes, Gwydir Wetlands and Macquarie Marshes, 12 wetlands of national or state significance ([21], Table S4 therein), and many others of ecological and cultural importance to Indigenous Peoples.

Floodplain harvesting involves capture of rainfall runoff and floodwater in large on-farm storages. In NSW, this water is an unlicensed, free bonus to licensed water entitlements and between a third and a half of irrigation water use is from floodplain harvesting

[52]. The NSW government is seeking to licence, monitor, and regulate the practice [53]. The Department of Planning Industry and Environment (DPIE) acknowledges the take exceeds legal limits under the Basin Plan ([54], p. 12). Growth in storages increased 2.6-fold in 26 years (557 GL in 1993–94 to 1393 GL in 2019–20) [52,53], and the average take was estimated at 778 GL/year, twice the volume modelled by DPIE [52].

In the Barwon–Baaka/Darling River, floodplain harvesting has contributed to increased cease-to-flow events, depleted baseflows, previously near-annual flow pulses reduced in frequency by >90%, and loss of biodiversity and ecosystem integrity [55]. In 018–19, the year of catastrophic fish kills at Menindee, an estimated 845–1135 GL irrigated the northern NSW cotton crop, of which ca. 1000 GL evaporated from storages before use, but only 40 GL flowed past Bourke, and 11 GL reached Wilcannia ([56], p. 3). Domestic water supply, sourced from the river, failed to meet minimum quality standards [57]. Yet, access to safe, reliable drinking water is a basic human right [58]. Governments of member States bear the duty of providing safe water to their citizens. Accordingly, we consider floodplain harvesting is a major water justice issue, not only a water management issue.

### 4.3. The Water Market—Increasing the Gap between the "Haves" and "Have Nots"

Commodification and trading of water was envisaged under the NWI as a means of ensuring water moved to the highest value productive use [9]. However, due to a lack of regulation and poor design, the market became a complex, volatile, and opaque trading system, which requires detailed understanding of rules and regulations, transactions data and trading products. In this regard, smaller, family-based irrigators trading on the temporary market to support farm income are losing out in trying to compete with far more sophisticated investors, such as international agricultural corporations and institutional financial traders [59,60].

High prices for temporary water, transparency, and fairness were identified as common challenges faced by irrigators when trading water ([61,62], p. 46). The Australian Competition and Consumer Commission (ACCC) Inquiry into Murray–Darling Basin water markets found serious undermining in trust of markets: "many water users reported that they do not trust that the markets and key institutions are fair or working to the benefit of water users, in particular irrigation farmers" ([63], p. 2).

## 5. The Implementation Gap

### 5.1. Regulatory Capture

Regulatory capture is public decision-making that favours particular interests and stakeholders over the broader public interest [64]. The model of vested interests capturing the political and regulatory systems to control resources has a long history in the Basin, commencing with the rise in power in the 1840s and 50s of the squatters—the holders of vast pastoral leases [1]. This model continues today with the political influence of irrigator lobby groups backed by large corporate agricultural companies [8].

Regulatory capture was highlighted by the NSW Independent Commission Against Corruption (ICAC) in November 2020 when it found water policies had been undermined by bureaucrats who has been "manifestly partial" towards the irrigation industry [65]. ICAC concluded that "certain decisions and approaches taken by the department with responsibility for water management in NSW over the last decade were inconsistent with the object, principles, and duties of the WMA (*Water Management Act*, NSW 2000) and failed to give effect to legislated priorities for water sharing" ([66], p. 8). Recommendations included that "clear and transparent processes, underpinned by independent scientific studies, should be used to determine NSW Government's overarching water policy discussion" ([66], p. 147).

In another example, e-mails released in a call for papers by the NSW Select Committee on Floodplain Harvesting revealed DPIE officials had altered the modelling underpinning the Murray–Darling Basin Cap to allow for increases in irrigation take that were well

above the Cap in order to facilitate the licencing of floodplain harvesting [67]. DPIE used new modelling to increase Baseline Diversion Limits and SDLS to ensure floodplain harvesting take was within the Cap and then mischaracterised their "adjusted Cap" models as updates of official Cap and SDL models. DPIE's internal communications revealed their misgivings that the modelling was uncertain and that they did not fully understand many of the assumptions [68]. Yet, all decisions about compliance with Water Sharing Plans, Water Resource Plans, and SDLs are based on this modelling.

### *5.2. Compliance*

The capacity of governments to police rules on water reform has been negligible [69], and compliance provisions of the *Water Act* are inadequate [70]. The remedy was the establishment of the Office of the Inspector General of Water Compliance (OIGWC), nine years after Basin Plan implementation, with an office of about 30 staff and a part-time Inspector General. The OIGWC is a political fix for a conflict of interest, whereby the MDBA was responsible for both implementation and compliance [71]. The OIGWC role is political, not legal: "…to rebuild trust and confidence in the…Plan by providing communities the assurances they are seeking through strong and independent regulation of water compliance" [72]. The first inquiry announced by the OIGWC was not, as one might expect, into extensive, systematic non-compliance with the Basin Plan by the NSW Liberal-National government [15,73,74] but into the CEWO operations "due to mistrust in environmental water" among Basin communities [75]. It remains uncertain if OIGTW can influence the extremely low level of trust in governments and bureaucrats among Basin communities.

## 6. Policy Options for Change

Having outlined the gaps, we detail policy options to reconcile environmental water requirements with water scarcity as follows: (1) improved modelling and measurement of water availability and use; (2) regulating inflow interception activities; (3) more effective use of environmental water; (4) environmental triage and adaptation pathways; (5) addressing water injustice; and (6) transparent, accountable water governance. These options require re-framing the competing interests of water users to focus on water as a resource for adaptation.

### *6.1. Improved Modelling and Measurement*

The shortcoming in water accounting at Basin-scale was illustrated by the 22% difference between actual and expected river flows [31] and the incapacity of government agencies to quantify losses, including those due to surface-groundwater interactions, return flows, and floodplain harvesting. To improve this situation, moving from "single entry" water accounting (which only counts diversions) to "double entry" (which includes how much water remains in rivers) will enable discrepancies to be identified and models refined. The current Basin hydrological model is an out-of-date collation of unpublished, poorly ground-truthed State agency models. Keeping models secret enables political manipulation, as described above. In May 2021, the Federal Government allocated funding for a new hydrological model. It needs to be publicly accessible and independently verified so that its underlying assumptions and performance can be assessed openly and transparently.

Independent monitoring and reporting of Basin sustainability indicators are needed. Since Basin governments de-funded the Sustainable Rivers Audit [5], the only public reporting has been the 2020 Basin Plan Evaluation [12] and reports by CEWO. The former lacks independent measures of outcomes and impacts and is another example of the MBDB marking its own homework. The CEWO Long-Term Intervention Monitoring program (now Flow MER) only tracks changes for a small proportion of wetlands that receive Commonwealth environmental water and does not report on the condition of the entire

6.3 million hectares of basin wetlands, reporting misleading outcomes instead, such as a wetland was "influenced" by environmental water when only a small part of it was actually inundated [21].

### 6.2. Regulating Inflow Interception Activities

A cap-and-trade water market, as in the Basin, only works if water entitlements are actively policed. The egregious example of floodplain harvesting demonstrates the risks to environmental and socio-economic values if some users can take water at will. Water intercepted by farm dams, floodplain harvesting, reduced return flows, groundwater extraction, and increased transpiration from afforestation and regrowth after fires need to be quantified to inform reviews of the *Water Act* and Basin Plan in coming years. Some of these activities, like floodplain harvesting, should be capped and regulated as soon as possible. Others require detailed consideration of possible policy interventions, such as how to decommission small farm dams at a scale that will actually increase river inflows. In the case of afforestation, the trade-offs between benefits and water losses require societal value judgements. Options include limits on location, timing, and species for forestry plantations and requiring owners to acquire water entitlements to off-set increased transpiration [76].

### 6.3. More Effective Use of Environmental Water

Conservation of freshwater biodiversity in the Basin can be enhanced by better management of environmental water, including timing of releases to ensure evaporative loss is minimised [21]. The limited environmental water available could sustain more wetlands if released from dams in pulses that fill river channels and are allowed to spill onto floodplains and private land. Basin governments agreed in 2013 to relax so-called "flow constraints" by 2024 in seven river reaches by purchasing flood easements from 3300 affected landowners, relocating or strengthening roads and bridges and realigning flood control infrastructure [48,77]. In the southern Basin, constraints management would conserve an extra 375,000 hectares of floodplain wetlands. To date, State governments have made no progress in implementing their commitments.

Benefits of environmental flows are greatly diminished by the release of unnaturally cold water from the base of large headwater dams, preventing breeding of native fishes for up to 300 km downstream as well as the many weirs that lack fish passage structures and prevent spawning migrations [78]. Infrastructure modifications could largely eliminate these negative impacts.

### 6.4. Environmental Triage and Adaptation Pathways

In view of diminishing water availability, a triage framework is needed whereby environmental and socio-economic objectives are defined, agreed, prioritised, and implemented for each catchment, establishing an adaptation pathway approach [79,80]. Triage decisions are already made by default as environmental water managers choose which wetlands to prioritise; less than 2% of the "managed floodplain" receives environmental water annually [21]. Triage decisions need to be made further in advance, with greater quantification and transparency, using adaptation pathways planning processes.

The Wentworth Group proposed an adaptation pathway approach based on two sets of triggers to maintain environmental and socio-economic values [81]. Firstly, for triggers and thresholds for wetland values, triage decisions need to be based on the following conservation principles: (1) prioritising viable populations of threatened and migratory species; (2) maintaining ecological character of Ramsar wetlands according to Australia's international commitments; (3) representative conservation of distinct freshwater ecological communities; and (4) maintaining water quality to avoid risks to wildlife and people. Triggers need to be defined in water plans to balance environmental and consumptive water use. Environmental and socio-economic thresholds, once crossed, would then result in

triage decisions to rebalance environmental and consumptive water use. The second set of triggers is based on river flow targets representing priority water uses, such as town water supplies and important wetlands. Flow targets would be linked to water access rules to only allow upstream irrigation diversions once the flow targets are achieved. The two sets of triggers need to be statutory and linked to specific actions to avoid governments procrastination on "pulling the trigger."

A further option is annual adjustment of the volume of the entire entitlement pool according to reduced water availability [81], allowing for gradual changes rather than a large step change at the end of each water plan period. Water markets would enable trading of reduced entitlements to maximise economic benefits. Regular review of water management plans would enable adaptation of the triage framework. Triage processes must include structural adjustment and other measures to support communities to transform to a future with less water. Outcomes and priorities for each catchment are social and political decisions.

### 6.5. Addressing Water Injustice

Water reforms and governance in the Basin has resulted in communities of "haves" and "have nots." It is unjust that many communities have been unable to access water to meet critical human needs, including towns along the Baaka/Darling River, such as Wilcannia, Menindee, and Pooncarie, while water is diverted upstream for lower priority irrigation use. Downstream flow targets and other policy options proposed above would begin to improve water justice.

It is unjust that 40 Indigenous Nations in the Basin have experienced water dispossession and hold only 0.2% of water entitlements [82]. The NSW Government proposal to create floodplain harvesting licences and transfer all but one to non-Indigenous interests compounds this injustice [81]. Basin governments need to find meaningful, inclusive ways to increase water entitlements to meet cultural, environmental, and socio-economic needs of First Nations peoples.

### 6.6. Transparent, Accountable Water Governance

We have illustrated how reforms to reconcile environmental water requirements with scarcity are failing due to regulatory capture and poor compliance. It remains to be seen whether compliance will be improved by the recently established Federal Inspector-General of Water Compliance and NSW Natural Resources Access Regulator. However, their terms of reference do not address systematic governance failures. Accordingly, further governance reforms should be implemented, namely (1) establishing quantified water management targets for the environment and people; (2) requiring publicly-funded Basin research to be genuinely peer reviewed and published to academic standards; (3) making models and data open access; (4) reinstating independent review and monitoring processes; (5) enabling the Federal Government to hold Basin States to account for implementing their commitments, including withholding funding in the event of non-compliance; and (6) providing for third-party standing to enforce the *Water Act* and Basin Plan in the Federal Court.

### 7. Concluding Remarks

Norman Lindsay's classic Australian children's book *The Magic Pudding* is about Albert, the talking pudding (meat pie); no matter how much is cut from him, the gap magically closes so that the pudding may be eaten again [83]. Sadly, the Basin Plan is proving to be a similar fantasy, contending that everyone can have as much water as they need to farm and keep wetlands healthy, when in reality, hard trade-off decisions are unavoidable.

We have outlined the substantial gap between water requirements of flow-dependent ecosystems in the Basin and the water allocated to sustain them, the gap between

rhetoric of the Basin Plan and actual implementation, and the trust gap generated by poor governance. The policy options we have outlined provide for conserving freshwater biodiversity with less water. These options are pragmatic in accepting that extensive areas of wetlands will be lost, and society will need to make tough triage decisions between the environment and socio-economic interests. We argue that this decision making needs to consider water as a resource for cross-sectoral adaptation within a system of honest, transparent, and accountable governance.

Therefore, how can some of the experiences of implementing water reforms in the Basin be generalised to lessons for water reform programs elsewhere? Perhaps the most important learning from the implementation process to date is the imbalance between the considerable effort and resources expended on technical assessments, hydrological modelling, and scientific monitoring and evaluation and the essential but relatively underdone requirement to engage communities and stakeholders *as partners* who have an understanding and some degree of ownership of the reforms. The top-down, technocratic approach adopted by the implementing agencies has failed to convince affected communities that the purpose of the reforms is for the public good and essential to the management of water as a common pool resource. A much stronger emphasis on bottom-up, community-led co-production, visioning, and co-learning, integrated with top-down processes, would have provided a much stronger basis for an adaptive approach to implementation [84]. Such an approach would enable communities to better address current and future challenges, such as the pressing need to adapt to climate change.

Despite the plethora of laws, rules, and regulations, the Commonwealth and State governments have no effective control over the actions of irrigators located in regional and sometimes remote parts of the Basin far from the centres of power and decision-making. However, the public has also witnessed lack of compliance by some Basin States with their obligations, such as "the cavalier manner in which NSW is managing water including exploiting every loophole in the national water reforms to allow more water to be extracted for irrigation at the expense of the environment, Indigenous nations, and downstream communities and industries" [68].

In the lead-up to the review of the *Water Act* (due in 2024) and the Basin Plan (due in 2026), it is time that major reforms were put in place to prevent State governments evading their responsibilities under the Murray–Darling Basin Agreement.

**Supplementary Material:** The following supporting information can be downloaded at: https://www.mdpi.com/article/10.3390/w140202084/s1, Table S1. Environmental watering volumes (ML) used (held and planned environmental water) between 2007-08 and 2020-21.

**Author Contributions:** M.J.C. and J.P. both contributed equally for this manscript. All authors have read and agreed to the published version of the manuscript.

**Funding:** This research received no external funding.

**Acknowledgments:** We thank Brian Richter for inviting us to contribute to this special issue.

**Conflicts of Interest:** Jamie Pittock is a member, and Matt Colloff an associate, of the Wentworth Group of Concerned Scientists. The authors declare that they have no other conflicts of interest.

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
