# Peer review of "Mind the Gap! Reconciling Environmental Water Requirements with Scarcity in the Murray–Darling Basin, Australia"

_water, doi:10.3390/w14020208_

Round 1

Reviewer 2 Report

The authors have examined critical issues in the Murray-Darling basin planning and water management. They have presented policy options that can address the need to adapt to less water and re-frame the current policy context. The idea of the manuscript is good. I recommend the manuscript for publication after addressing minor revision described below.

Line 13 – What does ESLT stand for?

Line 13 – What does SDL stand for?

Line 108 – Authors are expected to mention the time frame for such reduction rate.

Line 110- What are the “best available science”? Please bring info/examples of such approach/strategy.

Line 228 – It is good to see that the potential impacts of climate change on the basin has been reviewed. However, authors are expected to come up with a strategy/solution/suggestion to address such climate change impact. What do authors suggest for addressing negative impacts of climate change in this basin?

Reviewer 3 Report

Many of the concepts and agencies introduced in this paper will not be familiar to non-Australian readers. The authors should provide a longer introduction that more thoroughly 'sets the stage' for this commentary on the Murray-Darling Basin. This includes carefully defining key concepts, legislative bodies and processes, and terminology that may not be familiar to non-Australians. I have noted many terms throughout the document that will require definition or a more thorough explanation of context. 

The paper provides a thorough and compelling critique of existing policies, laws, and practices, and also offers constructive, corrective measures.

Reviewer 4 Report

Dear Authors,

your paper is very interesting and well-written, and I think it could provide readers interested in the Murray–Darling Basin with very useful insights and a clear picture of the actual situation.

I have just a few very small comments:

  1. be consistent in following the Journal guidelines for the references (e.g., line 251);
  2. could you develop a bit the manuscript, showing if your results could be somehow used outside of the case study? Indeed, I am a bit concerned about the very local focus of the work (even if I acknowledge the great importance of the basin in Australia).

Round 2

Reviewer 3 Report

It appears that these authors have sufficiently responded to my comments.